# Walking Behaviour of Individuals with Intermittent Claudication Compared to Matched Controls in Different Locations: An Exploratory Study

**DOI:** 10.3390/ijerph20105816

**Published:** 2023-05-13

**Authors:** Anna M. J. Iveson, Ukachukwu O. Abaraogu, Philippa M. Dall, Malcolm H. Granat, Brian M. Ellis

**Affiliations:** 1Research Centre for Health, Glasgow Caledonian University, Glasgow G4 0BA, UKukachukwu.abaraogu@gcu.ac.uk (U.O.A.);; 2School of Health Sciences, Salford University, Salford M5 4WT, UK; m.h.granat@salford.ac.uk

**Keywords:** intermittent claudication, physical activity, walking, location, GPS

## Abstract

Individuals with intermittent claudication (IC) are less physically active than their peers, but how this varies with location is unclear. Individuals with IC and matched controls [sex, age ±5 years, home < 5 miles] wore an activity monitor (activPAL) and carried a GPS device (AMOD-AGL3080) for 7 days. GPS data categorised walking events as occurring at home (<=50 m from home co-ordinates) or away from home, and indoors (signal to noise ratio <= 212 dB) or outdoors. Number of walking events, walking duration, steps and cadence were compared between groups and each location pair using mixed model ANOVAs. In addition, the locus of activity (distance from home) at which walking was conducted was compared between groups. Participants (*n* = 56) were mostly male (64%) and aged 54–89 years. Individuals with IC spent significantly less time walking and took fewer steps than their matched controls at all locations, including at home. Participants spent more time and took more steps away from home than at home, but were similar when walking indoors and outdoors. The locus of activity was significantly smaller for individuals with IC, suggesting that it is not just physical capacity that influences walking behaviour, and other factors (e.g., social isolation) may play a role.

## 1. Introduction

Peripheral arterial disease (PAD) is atherosclerotic occlusion of arteries in the lower extremities affecting over 230 million individuals worldwide, with a prevalence of 5% in people aged 25 or over rising to 21% in older adults [1]. Although many patients with lower extremity PAD are asymptomatic, intermittent claudication (IC) is considered a key symptom, in which individuals experience cramp-like pain in their lower limb(s) during walking due to insufficient lower limb circulation to meet the metabolic demands of the muscles [2,3]. Supervised exercise therapy is one of the primary recommendations for the management of IC and PAD and has been shown to improve maximum and pain-free walking distances and quality of life [4,5]. However, there is often a lack of provision of supervised exercise therapy, and it can be burdensome for the patient and difficult to access [5,6]. Home-based exercise programmes, which provides structured advice on walking exercise and may include advice to increase physical activity (PA), often alongside other behavioural components, may offer a more accessible and preferable alternative to supervised exercise therapy. Home-based exercise programmes can be effective when monitoring of exercise is included, but additional research is required to identify effective components [5,7]. Whilst lower PA levels are a predictor of increased morbidity and mortality in individuals with PAD [8], walking confers several benefits including improving functional capacity and protecting against cardiovascular risk factors [9,10].

Individuals with IC have been consistently shown to have lower levels of physical activity when assessed with body-worn sensors, spending less time walking [11], taking fewer steps [11,12,13,14], and spending less total energy [12,15], compared to groups without PAD and IC matched for age and sex. Compared with matched control groups, individuals with IC have also been shown to walk with a lower average cadence [11,13], spend less of their time walking at higher cadences [11,13], and have a more broken up pattern of walking [14]. Barriers to walking in individuals with IC include lack of knowledge about their condition and expectations, poor walking capacity before having to stop due to pain, comorbidities and walking-induced pain [16]. Walking-induced pain can be modified by the environment, for example qualitative reports from individuals with IC indicate that cramp-like pain is particularly prevalent during long or strenuous exercise, such as walking uphill and climbing stairs [16,17,18]. Additionally, people with IC may modify their walking behaviour, for example by slowing down or stopping to avoid such pain [17].

One option for adding context and exploring the interaction of physical activity with the walking environment is the integration of physical activity monitors with global positioning system (GPS) devices [19,20], which use satellite tracking to record the location of individuals. GPS devices have been used in a range of different populations to provide context to device-based measures of physical activity, for example exploring use of wide-range of specific facilities, such as parks [21], or assessing where physical activity takes place [22]. American adults spent 47 min/day in moderate-to-vigorous physical activity, of which 19% was spent at home (<125 m), 41% close to home, and 40% further away from home (>1666 m), identified by distance from home buffers [23]. Whereas, when exploring location using visual inspection of Google Maps, American adults spent 20% of their time in moderate-to-vigorous physical activity at home, 28% on roads, and 13% in parks [24].

A number of studies have used GPS technology to study free-living walking adaptations in people with IC [25,26,27,28]. However, these studies have explored walking within a constrained environment and time period, for example over a fixed period of 45–60 min recorded in researcher-designated public parks. Frequently the aim is not to assess the wider environmental context of habitual walking in people with IC, but to provide a real-world assessment of physical capacity, and the recording locations were selected to be free of constraints such as hills and motorised vehicle traffic. Therefore, this does not fully represent the habitual walking in the daily life of individuals with IC in the community.

Whilst it is clear that the total volume of walking undertaken by people with IC is lower than people without IC, no previous study has explored the location of habitual walking in the daily life of individuals with IC in their community, nor has this been compared to the location of habitual walking of people without IC. Understanding where and how individuals with IC usually walk could be helpful in identifying opportunities and strategies to support individuals to increase physical activity, for example as part of a home-based exercise programme. Therefore, this study aims to compare the free-living walking characteristics of pattern (number of walking bouts), volume (duration of walking, number of steps taken) and intensity (cadence) of individuals with IC compared to age, sex and home location-matched controls undertaken in different locations. Specifically, we compared walking at home with walking away from home, and compared walking indoors with walking outdoors. Additionally, we aimed to compare the locus of walking (how far away from home walking took place) between individuals with IC and matched controls.

## 2. Materials and Methods

A case-control study was conducted using pairs of individuals with intermittent claudication (IC) and matched controls who were asked to wear an activity monitor and carry a GPS device for 7 days. Ethical approval for the study was granted by the NHS West of Scotland Research Ethics Committee (12/WS/0067). All participants gave verbal and written informed consent.

### 2.1. Participants

A convenience sample of individuals with intermittent claudication (IC) was recruited by sending study information out along with routine appointment details to returning patients from the Community Claudication Clinics (a specialist vascular screening service) across NHS Lanarkshire, UK. For this exploratory analysis, we did not conduct an a priori sample size calculation. Individuals were included in the study if they had a medical diagnosis of IC from the claudication clinic (based on the Edinburgh Claudication Questionnaire [29], confirming that pain was associated with walking and resolved with rest). Individuals were included if the principal condition limiting their walking was IC (participant self-report), and if they had not undergone surgical intervention for IC. Individuals were excluded if they had self-reported unstable angina pectoris, respiratory or neuromuscular disease or a history of stroke if these conditions had a detrimental impact on their walking ability. Therefore, participants with IC included in the study had Stage II Fontaine’s classification of IC [3]. Furthermore, individuals with a significant cognitive or physical impairment which may limit their ability to participate, or who had insufficient use of English or for logistical reasons could not participate, were excluded. Individuals with self-reported signs of rest pain, gangrene or tissue loss were also excluded.

Following recruitment of individuals with IC (between May 2012 and June 2013), matched control participants (CO) were recruited (between June 2013 and December 2013) from the general population in Lanarkshire, using a range of techniques including: e-mails to staff at Glasgow Caledonian University; posters put up at community locations around Lanarkshire (e.g., supermarkets, libraries, community centres); leaflets handed out at talks to lay groups; snowballing from other participants. Individuals were matched based on sex, an age within 5 years, and home address within 5 miles radius (to increase the likelihood that individuals would walk within a similar terrain and facilities). Control participants were screened and consequently excluded from the study for self-reported symptoms of IC or discomfort in their legs when walking.

### 2.2. Tools

Daily free-living walking was measured using the activPAL (the original uni-axial version), a small (53 mm × 35 mm × 7 mm), body-worn, uni-axial accelerometer-based physical activity monitor (PAL Technologies Ltd., Glasgow, UK). The activPAL device which weighs 20 g measures accelerations at 10 Hz from which individual positions are classified as either sitting/lying down, standing or stepping using proprietary software algorithms. The activPAL has been validated for differentiating standing, sitting and stepping, as well as cadence and step count in healthy adults [30,31,32,33] and older adults [34]. It has also been used to evaluate differences in PA measures in patients with IC and controls [11,14] allowing walking events (continuous periods of walking), defined as ‘event-based’ analysis [35] to be differentiated as a consequence of IC.

Location information was gathered using the AMOD AGL3080 (AMOD Technology Co., Ltd., Taipei, Taiwan) GPS device. AMOD AGL3080 is a pocket-sized (90 mm × 45 mm × 23 mm), 95 g weight (including batteries) device which communicates with GPS satellites to record its position every 5 s (providing it has a signal) and is capable of storing approximately 360 h (i.e., 15 days) of detailed logging. The GPS device has a positional accuracy for latitude and longitude of 10 m [36], although this will be dependent on environmental factors [37]. Preliminary testing prior to this study aggregating data from 10 walks over 3 days round a U.K. city indicated that mean difference for a static position (GPS device placed on Ordnance Survey Trigonometric point T072) was 1 m for latitude and longitude, and for stops during a walk (six locations with repeat position estimated by user) was within 5 m for both latitude and longitude [37]. The GPS device was carried in a coat pocket during these tests and the effect of wear location on positional accuracy was not explored. These values were similar to the previous literature reporting variability in GPS-recorded coordinates [38,39,40].

### 2.3. Protocol

Individuals attended a single research visit (day 0). Demographic information (including sex, age, address and postcode, self-reported height and weight, work status, and car ownership) was collected. Information on disease severity was obtained from medical notes rather than being specifically measured for the research study. All assessment of peripheral blood flow were taken by specialist staff working in a vascular screening clinic. In the clinics, disease severity could be evaluated using various metrics. Typically, the ankle-brachial index (ABI) was employed, which is the ratio of ankle to brachial systolic blood pressure. This measurement was taken while the participant was supine, either using a handheld Doppler device or machines provide simultaneous readings. Where patients had diabetes or non-compression of the vessels were suspected, the Lanarkshire oximetry index (LOI) was employed, which uses a pulse oximeter at the toe and finger as an alternative to a handheld Doppler device [41]. ABI and LOI are interpreted similarly, with a value of ≤0.9 considered indicative of PAD [5].

Monitoring equipment was also provided at the research visit on day 0. A waterproofed (nitrile sleeve, PAL Technologies Ltd., Glasgow, UK, wrapped with Opsite Flexifix, Smith & Nephew, Hoofddorp, The Netherlands) activPAL activity monitor was then attached to the anterior aspect of the mid-thigh, using a hypoallergenic double-sided adhesive pad (PAL Stickie, PAL Technologies Ltd.), and covered with a waterproof dressing (Opsite Flexifix). Individuals were asked to wear the monitor for the next seven days (days 1–7), removing it on the morning of the 8th day. The monitor was worn overnight and during water-based activities. Participants were provided with a paper diary to record any periods of non-wear of the activity monitor, we did not ask participants to record time in bed.

Individuals were also given the GPS device which they were instructed to switch on and carry with them during their waking hours, even when at home. The GPS did not have a belt clip and participants were asked to carry it in a pocket or bag. Participants were specifically asked to ensure they wore the device on trips out of the home. Participants were required to change the AAA batteries each day, and were provided with a battery charger and spare batteries to allow charging of the spares overnight. At the end of the monitoring period, participants returned the activPAL monitor and GPS device by post.

### 2.4. Data Processing

The data processing is described in detail elsewhere [37]. Briefly, data from the activPAL was downloaded using activPAL software v.5.9.1.1, using the events output (VANE algorithm) and then post-processed using custom software (HSC PAL Analysis Software v1.19s) to collate strides into walking events. We were interested in all walking, and did not impose a minimum duration on walking events. GPS data were downloaded onto a computer, converted into Ordnance Survey Grid co-ordinates and imported into ArcMap™ GIS software.

Diaries were used to identify any self-reported non-wear periods of the activPAL, and additional visual inspection of the data was performed in the proprietary activPAL software to remove days with obvious non-wear (for example very long periods of continuous standing, or very low step count). As we were interested in assessing walking, we did not attempt to isolate waking hours and activPAL data were considered to be continuous for 24 h, unless non-wear was reported or identified through visual inspection of the data. There was no lower limit set on duration of GPS for a valid day of data. Data from a participant were included in analysis if there were at least 4 days with concurrent activPAL and GPS data for assessment. Longer data collection periods (e.g., 14 days) are required to adequately capture some types of activity space use [42,43]. However, for broad usage categories of activity space such as indoors and outdoors, this threshold was selected pragmatically to provide a reasonable snapshot of usual behaviour.

#### 2.4.1. Defining Location-Based Parameters

The home location of each participant was determined using the mapping software ArcGIS™. A MasterMap^®^ layer was displayed in the mapping software showing aerial view outlines of buildings. The Eastings and Northings of the centre-point of the home perimeter (determined by the researcher) were defined as the home location.

A Euclidean buffer [44] with a threshold of 50 m distance from the home coordinates was used to distinguish between walking activity occurring ‘at home’ and ‘away from home’. The value of 50 m was selected because previous studies have used this threshold as either a buffer around each participant’s home to remove points from analysis [21] or as a threshold for including activity at or close proximity to home and workplace [45]. Although many premises are less than 50 m in extent, errors due to being indoors indicate that using a buffer zone of 50 m for collapsing data to the home location may be advisable.

The signal to noise ratio (SNR, the ratio of signal power to the background noise), was used to distinguish between indoors and outdoors locations. SNR is used in some web-based applications as a way of determining indoor from outdoor data points (e.g., PALMS, currently hosted at HABITUS https://www.sdu.dk/en/habitus, accessed on 16 March 2023). In this study indoors was defined as ≤212 dB, and therefore outdoors was >212 dB [46,47]. The feasibility of using this threshold value was verified by graphing the SNR values along with the activPAL walking events, GPS distance from home and diary data for whether they reported being away from home.

#### 2.4.2. Integration of GPS-Derived Outcome Measures with Walking Events

All walking events were categorised as taking place either at home or away from home, and separately as taking place either indoors or outdoors, based on their correspondence with GPS data points, using the methodological processes described in detail in Iveson et al., 2020 [37]. Distance during each walking event was calculated as the sum of the Euclidean distance between successive pairs of GPS points in that event. The locus of activity, describing the area around an individual’s home where their activity occurs, was calculated for each walking event using the Euclidean distance of GPS points during the walking event from the home location. It should be noted that the location of walking might have been reached by walking or by other modes of transport such as by car, especially for larger distances. Therefore, this outcome describes the area in which the person operated, as opposed to representing how far they walked from their home.

#### 2.4.3. Physical Activity Outcome Measures

Outcome measures were calculated for each day for each participant, and presented as a daily average. Four physical activity outcome measures were derived using only data from the activPAL activity monitor, number of walking events, walking duration, number of steps, and average cadence (mean value of all walking events), which were reported for all locations. Additionally, three physical activity outcome measures were derived using integrated activPAL and GPS data, distance walked, average distance from home and maximum distance from home. These were reported only when the walking took place both away from home and outdoors. GPS data are less reliable when measured indoors and outcomes such as distance between successive GPS points may not represent the actual distance walked when moving around a building.

### 2.5. Statistical Analyses

At any point in time, the location of an individual will be categorised based on both whether they are home or away from home and whether they are indoors or outdoors, leading to four mutually exclusive locations: indoors at home, outdoors at home, indoors away from home and outdoors away from home. Although it assessed walking twice, in this analysis, we have explored each pair of locations (home versus away from home and indoors versus outdoors) in separate models. This allows an initial exploration of broad categories of location, as opposed to small silos. It also allows for comparison with other work, which has explored differences in home versus away from home locations in other populations. Following this, we have assessed walking behaviour in one of the mutually exclusive locations, outdoors away from home. This allows exploration of walking metrics derived from GPS data.

Statistical analysis was carried out using IBM SPSS v.26, and a two-sided *p*-value of 0.05 was used to assess significance. Mixed methods ANOVAs were used to compare physical activity outcome measures between individuals with IC and their matched controls (between-subjects factor) for pairs of specified locations (within-subjects factor). Separate models were conducted for home vs. away and indoors vs. outdoors. For the single location of being both away from home and outdoors, independent samples *t*-tests were used to compare physical activity and GPS-based outcome measures between individuals with IC and their matched controls. Model assumptions were tested using the Shapiro–Wilks test for normality, Levene’s test for equality of error variances and Box’s M test of equality of covariance. Where assumptions were not met, data transformation was attempted using square root or cube root. When transformation was unsuccessful, a Mann–Whitney U test was conducted instead of the independent sample *t*-test.

## 3. Results

Thirty-five individuals with intermittent claudication (IC), and thirty-four matched control (CO) participants were recruited (recruitment of individuals with IC was undertaken first and we were unable to find a suitable control match for one participant with IC within the study time period). Data were analysed from 28 pairs of participants (56 individuals) with at least 4 days of data. All participants had seven days of physical activity data, and thirty-eight participants had seven days of GPS data (mean 6.5 ± 0.8), covering a mean of 10.5 ± 2.5 h/day. Participants were mostly male (*n* = 18 men, 64%). Both groups of participants had a mean age of 67 years (IC range: 55–89 years; CO range: 54–86 years). Participants with IC and the control participants had a similar mean BMI (calculated from self-reported height and weight) of 26.84 kg/m^2^ and 27.02 kg/m^2^, respectively, which is considered overweight (IC range: 18.07–43.23 kg/m^2^; CO range: 20.50–38.45 kg/m^2^). Participants were all successfully matched for sex, age and distance between home (<5 miles). There was no difference in BMI between participants with IC and controls (paired *t*-test; *p* = 0.86). The mean ABI of participants with IC was 0.71 (*n* = 20; range: 0.54–1.12) and the LOI was 0.78 (*n =* 7; range: 0.58–1.14), data was missing for one participant. As a whole, the group with IC had moderate disease severity, but it should be noted that some ABI (*n =* 3) and LOI (*n =* 1) values were higher than 0.9.

Three outcomes (duration of walking, total steps and average cadence) were successfully square root transformed prior to analysis (Table 1 and Table 2). The number of walking events was successfully cube-root transformed for the indoors versus outdoors comparison prior to analysis, but transformation was not successful for the home versus away comparison. In the absence of a suitable non-parametric equivalent test, results for this comparison have been presented with cube root transformation, but should be treated with caution. When walking in locations which were both away from home and outdoors, three outcomes (duration of walking, total steps and distance of walking) were successfully square root transformed prior to analysis (Table 3). Three outcomes (number of walking events, average distance from home and maximum distance from home) were not successfully transformed and are reported using the Mann–Whitney U test.

Across all locations (home versus away, Table 1; and indoors versus outdoors, Table 2), individuals with IC walked less than controls, taking significantly fewer walking events, for a significantly shorter duration of walking and taking significantly fewer steps. However, there was no difference in the average cadence of walking between the groups of participants.

Participants took a significantly smaller number of walking events away from home than at home (Table 1). However, although there were fewer walking events taken, participants spent significantly longer walking, took significantly more steps and walked at a significantly higher cadence away from home than at home. Both groups took significantly fewer walking events outdoors than indoors (Table 2). However, there were no differences in duration of walking and number of steps taken indoors compared with outdoors. Participants walked with a significantly higher cadence outdoors compared with indoors. The interaction between participant group and location was not significant for any of the outcome measures or locations.

The number of walking events and the cadence were not significantly different between the groups (Table 3). However, the individuals with claudication spent significantly less time, walked less distance and took fewer steps when walking away from home and outdoors than the controls. Each day, individuals with IC on the average spent 36% less time walking away from home and outdoors, walked 50% less distance and took 47% fewer steps than the control participants. Both the average and the maximum distance from home where walking occurred were significantly lower in the individuals with IC compared to controls. However, it should be noted there was large variability in both outcome measures.

## 4. Discussion

In this study we concurrently measured device-based walking and location across a week in their usual habitual activity and free-living environment in a group of people with IC compared with an age, sex and home location-matched control group. As expected, the total free-living physical activity of the individuals with IC was lower (fewer walking events, less time spent walking and fewer steps taken) than the controls, although they appeared to walk at a similar intensity (cadence). Individuals with IC habitually walked about half the distance and took about half the number of steps compared to the matched controls. This difference in physical activity between groups was consistent across all locations. Participants walked for significantly longer and took more steps and walked with a higher cadence, but in fewer events when walking away from home compared with at home. Participants also walked in fewer events and at a higher cadence outdoors compared with indoors, but the duration and steps taken were similar. The current study also found that people with IC walked with a considerably smaller locus of activity than the control participants. As locus of activity is higher than actual distance walked, this finding is not just about how much people walk, but where they do it, suggesting that the people with IC were more limited in their use of the environment.

Consistent with other studies [11,12,13,14,15], the current study found that people with IC had a significantly smaller volume of physical activity compared with a matched control group. But when comparing the actual volumes undertaken, the people in the current study appeared to do less than those in other studies. Our group of people with IC took 3524 ± 1655 steps/day, compared with 4737 ± 2712 [12], 6298 ± 3114 (converted from strides) [13] and 6524 ± 2710 [11,14]. Similarly, they walked for a shorter duration (44 ± 20 min compared with 90 ± 36 min [14]) and had fewer walking events (160.1 ± 98.9 compared with 415.0 ± 160 [14]). However, the average cadence of walking was more similar (72.6 ± 5.8 compared with 68.2 ± 5.6 [14]). Although other studies have reported cadence as an outcome, the metrics used are not comparable with those used in the current study [13,48,49]. It should be noted that the volume of physical activity undertaken by the control participants was also lower in the current study, for example taking 6645 ± 2964 steps/day compared with 8672 ± 4235 [12], 8460 ± 3416 (converted from strides) [13] and 8664 ± 3110 [11,14]. It is unclear why this lower volume of activity occurred, as the participants appeared to be similar in age, sex, BMI and disease severity to the participants in comparable studies, and one comparable study [14] was conducted in the same geographical region (Scotland, UK) and using the same type of activity monitor (activPAL).

When looking at total duration of walking and number of steps taken (measures of volume of activity), individuals with IC consistently walked less than the control group at all locations. This implies that the difference in total physical activity was distributed across all locations, including when they were at home. Limitations to walking due to IC are more pronounced when undertaking longer periods of continuous walking [11,14], and may be assumed to be more pronounced when walking out of the home. Within the home location, there is not as much scope to take long continuous periods of walking [50], and it is therefore perhaps surprising that people with IC walked less than the control participants at home. This perhaps indicates the reduction in walking is related to expectations and self-efficacy, and not just purely a physical limitation. As well as measures of volume, participants with IC generally took fewer walking events than matched control participants. However, when walking both away from home and outdoors, the number of walking events was not statistically significant between groups. This implies that, in this location, people with IC were increasing the duration and number of steps within each walking event.

The current study found that individuals in both groups spent less time walking at home (IC 38%, controls 36%) than away from home. American adults have been reported to spend 10% [45] or 20% [23,24] of their time in moderate-to-vigorous physical activity at home. These studies used different methods of identifying the home location from GPS data, a 50 m threshold [45] which is equivalent to the current study, visually identified on a map [24], or a larger threshold (<125 m [23]). The higher amount of time spent at home in the current study, could be due to an older population, but could also be due to the difference in intensity of physical activity reported. All intensities of walking were reported in the current study, and it might be anticipated that lower intensity activity was more likely to be undertaken at home. This is supported by the fact that in other studies the proportion of time spent at home in activity increased as activity intensity was reduced, 30% of moderate to vigorous activity when categorised using a lower threshold (760 counts per minute vs. 2020 counts per minute [24]) and 56% of physical activity categorised as low [23]. Although people in the current study spent more time walking (IC 62%; control 64%) and took more steps (IC 68%; control 72%) away from home than at home, they did this in fewer walking events (IC 39%; controls 35%). This implies that the pattern of walking was different when away from home, taking place in longer continuous periods of walking. The walking also took place at a higher intensity (and likely a faster speed) when away from home than at home. This adds to the picture of more purposeful walking occurring away from home.

Considering the differences in indoor and outdoor walking behaviour, there were no differences in the time spent walking or number of steps taken between these locations in both groups, but there was a reduction in the number of walking events outdoors. Participants also increased their cadence when walking outdoors. There is some overlap when considering differences walking indoors and outdoors and home and away from home. Time at home will predominantly be indoors, although time spent in a garden or on very local streets might fall within the 50 m buffer of the home address. However, it is clear that not all time spent away from home is outdoors, as people may spend time in shops, at work or visiting other people. The pattern of walking indoors compared to outdoors further reinforces the view that it is walking both when away from home and outdoors that allows more purposeful, higher intensity walking. One study explored the free-living walking of individuals with IC using an ActiGraph activity monitor worn at the waist matched pain due to walking (time-stamped when participants pressed a button) to walking behaviour [51]. Participants only recorded a low number of instances of pain during walking or stopping from that pain, most of which occurred when walking outside their home (based on self-reported diary). The intensity of physical activity in these participants was also low, with only 8% of time spent in moderate-to-vigorous physical activity [51]. Taken together, this might imply that patients with IC will avoid activities further away from home, and when walking outdoors and away from home will compensate the need for frequent stops due to claudication pain by taking fewer steps, spending less time walking, and not walking to far distances. However, further research is required to understand whether this is purely imposed by limitations in physical capacity and pain onset, or if there are other or additional implications in learned response or self-efficacy.

Individuals with IC walked within a significantly smaller locus of activity (average 3.7 ± 4.3 km, maximum 5.5 ± 5.5 km) than control participants (average 7.2 ± 7.5 km, maximum 11.0 ± 10.0 km), indeed the group maximum distance from home of people with IC was lower than the group average distance from home of all walking in the control participants. These values do not represent distances walked from home, but may also include walking at a location which was reached by other transport, for example the car or public transport. In the current study, car ownership was slightly lower in individuals with IC (71%) compared to control participants (86%), which does not seem to explain the large difference in locus of activity. Thus, locus of activity represents the social environmental horizon of the individual, and not just their level of physical activity. Much GPS-based research explores differences between location using buffer distances, or exploring time spent at specific types of location. The exploration of how far from home physical activity is undertaken is less well explored. In one study, the median and IQR of distance from home of outdoor physical activity (light, moderate and vigorous) of UK adults was reported across various population characteristics, but locus was lower than reported here [52]. Typically, median distance from home was 0.6 to 2 km with an upper quartile 2 to 6 km, although for rural participants the median distance from home was 3.5 km and the upper quartile 15 km. In American adults, the distance from home to locations where moderate-to-vigorous activity occurred was 7.4 ± 17.4 km [45], which is similar to the control participants in the current study.

Reduced locus of activity in IC will have important implication for social participation and isolation in individuals with PAD and IC. Isolation has a strong independent associations to unrecognised PAD [53], and qualitative evidence indicates that a “feeling of isolation” is common in people with IC [18]. Future research may be needed to further explore the relationship between reduced locus of activity and having a diagnosis of IC; and the effect of PA interventions on locus of activity and quality of life. However, it appears there is a need for clinical intervention recommendations to encourage walking both away from home and outdoors to maximise potential for enhanced PA in IC. Such recommendations could also encourage patients to aim to enhance their locus of activity.

The main strength of this study was the use of concurrent measurement of physical activity and location of individuals with IC when undertaking their own habitual free-living physical activity for a week. Although a week can provide a snap-shot of a person’s physical activity, it may not represent their usual activity. To accurately identify a full-range of interactions with specific environmental features (for example commercial properties), a longer period of collection of GPS data may be required [42,43]. This is likely to be particularly problematic for the assessment of locus of activity as some journeys further from home may be uncommon and infrequent activities, which may simply have been absent on the week of measurement. Additionally, it was not practical to test participants with IC and their matched controls during the same or consecutive weeks. Differences in seasonal and weather conditions, such as hours of sunshine or precipitation might have affected their physical activity levels [54]. Although we matched individuals so that their home locations were within 5 miles of each other, it is possible that terrain and access to services such as shops or public transport were different between participants. Participants in the IC group had received a diagnosis of IC from the NHS specialist vascular clinics which they were attending, but we did not specifically measure any indicators of PAD or IC for this research project. Some of the measures of disease severity we obtained from medical records (ABI and LOI) indicated values which were in the normal range (>1.0). Equally, although we screened for symptoms of IC in the control group, it is possible that some individuals in that group had asymptomatic PAD. However, any misallocation to group would be anticipated to act in a conservative manner to reduce differences between groups. This analysis is an exploratory analysis, and we did not conduct an a priori sample size calculation. Results should therefore be viewed with caution, in particular it is possible that actual differences exist which we did not have the statistical power to detect. Although the positional accuracy of the GPS device was reasonable, we did not explore how such errors might contribute to the calculation of the distance outcome measure. Errors could be compounded in calculation of distance if positional errors in successive GPS points were not in a consistent direction. Additionally, the straight-line distance between GPS points may not represent actual path travelled. GPS devices can be used to explore transportation modes, and data from the more recent (tri-axial) versions of the activPAL monitor can be processed using the CREA algorithm which provides an assessment of seated transport. An assessment of transport alongside walking behaviour could provide additional information to explore the implications of locus of activity in this population, however that was beyond the scope of the current analysis. Finally, there is limited consensus of thresholds used to identify the home location and to separate indoor and outdoor activity. We used a SNR threshold of 212 dB to distinguish indoors and outdoors, based on a small (*n =* 8) validation study in adults [47]. The 50 m buffer used to identify the home location is used in other studies, but is likely to be larger than the individual building size of many homes, and some of the at home walking may have included some outdoor space, e.g., time spent in the garden. We only explored physical activity data that had concurrent GPS data available, which was 76% of time spent walking [37]. The activPAL activity monitors were worn continuously, but the GPS devices were switched off at times (e.g., when switching batteries), or may have been unable to locate a satellite (e.g., due to dense tree cover or some urban environments [38,39,40]).

## 5. Conclusions

The current study explored walking activity occurring at different locations, either at home or away from home and either indoors or outdoors, by integrating data from concurrent measurement of physical activity (activity monitor) and location (GPS). Individuals with IC spent less time walking and took fewer steps compared to matched control participants in total and across all of the explored locations, including when at home. Both groups spent more time and took more steps walking at home compared with away from home, but there was no difference between walking indoors and walking outdoors. As well as having a lower volume of physical activity, the locus of activity assessed by distance from home of walking activity was significantly lower for people with IC. This suggests that both physical capacity and psychosocial factors may influence habitual physical activity for individuals with PAD and IC.

## Figures and Tables

**Table 1 ijerph-20-05816-t001:** Walking events occurring at home and away from home for individuals with intermittent claudication and matched controls.

Outcome	Home		Away from Home		Between Subject Effect	Within Subject Effect	Interaction
	IC	CO	IC	CO	IC vs. CO	Home vs. Away	
Walking events (#) ^a^	97.25 ± 87.33	151.47 ± 85.55	62.88 ± 42.56	82.75 ± 59.60	F = 12.956 *p* = <0.001	F = 6.425 *p* = 0.014	F = 1.174 *p* = 0.283
Duration of walking (min) ^b^	16.61 ± 14.29	27.48 ± 14.40	27.08 ± 15.22	48.66 ± 28.08	F = 26.857 *p* < 0.001	F = 18.877 *p* < 0.001	F = 0.363 *p* = 0.549
Total Steps (#) ^b^	1135 ± 930	1886 ± 966	2390 ± 1475	4759 ± 2830	F = 28.970 *p* < 0.001	F = 44.368 *p* < 0.001	F = 2.122 *p* = 0.151
Cadence (steps/min) ^b^	69.24 ± 5.58	69.96 ± 5.31	76.12 ± 5.68	78.09 ± 7.31	F = 1.128 *p* = 0.293	F = 55.136 *p* < 0.001	F = 0.305 *p* = 0.583

IC = intermittent claudication group, CO = control group, values are shown as mean ± standard deviation. ^a^ Cube root transformed prior to analysis, transformation was not successful. ^b^ Data square root transformed prior to analysis.

**Table 2 ijerph-20-05816-t002:** Walking events occurring indoors and outdoors for individuals with intermittent claudication and matched controls.

Outcome	Indoors		Outdoors		Between Subject Effect	Within Subject Effect	Interaction
	IC	CO	IC	CO	IC vs. CO	Indoors vs. Outdoors	
Walking events (#) ^a^	112.29 ± 71.58	159.13 ± 56.98	47.84 ± 44.49	75.09 ± 42.65	F = 12.629 *p* < 0.001	F = 108.114 *p* < 0.001	F = 0.017 *p* = 0.896
Duration of walking (min) ^b^	24.11 ± 12.18	37.56 ± 14.22	19.57 ± 12.14	38.57 ± 23.57	F = 24.074 *p* < 0.001	F = 2.203 *p* = 0.144	F = 1.205 *p* = 0.277
Total Steps (#) ^b^	1815 ± 849	2917 ± 1181	1710 ± 1126	3728 ± 2580	F = 25.291 *p* < 0.001	F = 0.215 *p* = 0.645	F = 2.577 *p* = 0.114
Cadence (steps/min) ^b^	71.56 ± 5.19	70.35 ± 4.29	76.32 ± 6.81	75.47 ± 7.33	F = 0.518 *p* = 0.475	F = 50.359 *p* < 0.001	F = 0.062 *p* = 0.804

IC = intermittent claudication group, CO = control group, values are shown as mean ± standard deviation. ^a^ Data cube root transformed prior to analysis. ^b^ Data square root transformed prior to analysis.

**Table 3 ijerph-20-05816-t003:** Walking events occurring both away from home and outdoors for individuals with intermittent claudication and matched controls.

Outcome	IC Group	CO Group	Comparison
Walking events (#) ^a^	24.45 ± 20.12	30.68 ± 26.25	U = 1.131 *p* = 0.372
Duration of walking (min) ^b,c^	15.03 ± 10.50	29.58 ± 22.51	t = 3.102 *p* = 0.003
Total Steps (#) ^b,c^	1387 ± 1040	3083 ± 2432	t = 3.522 *p* < 0.001
Cadence (steps/min) ^b^	79.61 ± 6.99	83.29 ± 8.32	t = 1.791 *p* = 0.079
Distance walked (m) ^b,c^	1055 ± 676	2241 ± 1672	t = 3.459 *p* < 0.001
Average distance from home (m) ^a^	3714 ± 4308	7241 ± 7536	U = 216 *p* = 0.004
Maximum distance from home (m) ^a^	5479 ± 5453	11,030 ± 9967	U = 199 *p* = 0.002

IC = intermittent claudication group, CO = control group, values are shown as mean ± standard deviation. ^a^ Assessed using Mann–Whitney U test. ^b^ Assessed using independent samples *t*-test. ^c^ data square root transformed prior to analysis.

## Data Availability

The datasets used and analysed during the current study are available from the corresponding author on reasonable request.

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
