# Peer review of "Walking Behaviour of Individuals with Intermittent Claudication Compared to Matched Controls in Different Locations: An Exploratory Study"

_ijerph, 2023, doi:10.3390/ijerph20105816_

Round 1

Reviewer 1 Report

General comments

The present study explored the location of habitual walking in the daily life of individuals with intermittent claudication (IC) in their community and in comparison to matched controls.

This study is of interest considering the impact of ischemic pain on the daily walking behaviour of people with IC due to lower extremity peripheral artery disease. Such study is a real opportunity to provide objective data on activity location and better understand barriers or enablers to walking, and thus incorporate such information to design efficient home-based walking interventions.

The study is well written and clearly presented overall.

I have several comments below, including major comments regarding the definition of the population (and the participants inclusion), the statistical approach and some interpretations of the results. Major comments are highlighted with the symbol “*”.

Detailed comments

Introduction section

1.      **Line 25. The reported prevalence of IC is very low, the mentioned reference being outdated [i.e., ref #2, since ref #1 is not specific to the study of IC prevalence; also consider using only references related to the statements mentioned]. An important issue when reporting IC prevalence, relates to the definition of IC that is used. The prevalence of IC significantly differs according to the way it is defined. The definition of IC used by the authors is not mentioned in the manuscript. Once the definition of IC is specified, prevalence estimations should be aligned accordingly, as well as the assessment tool used during inclusion (see below, comment #6).

2.      Line 26. Would you mean “lower extremity peripheral artery disease”?

3.      Lines 28-29. Would you mean “supervised exercise therapy” when stating that “regular physical activity” is one of the primary recommendations for LEPAD management? At the conceptual level but also at the interventional level, this is not the same.

4.      Lines 32-33. Did the references #5 and #6 show the effect of walking on the development of a collateral blood supply? Again, please align the references with the outlined statements.

5.      *Lines 68-70. We understand that no previous study has explored the location of habitual walking in the daily life of people with IC. However, what such study would provide in the management of people with LEPAD and IC is not enough highlighted. Please elaborate to strengthen the interest of the present study.

Materials and Methods

6.      ***Participants. We have no information regarding how the ankle-brachial index (ABI) was measured and used to define LEPAD. Later in the results section of the manuscript (lines 274-275), we learn that ABI ranged between 0.54 and 1.12 (which means that ABI was measured). Considering that LEPAD is defined by an ABI ≤0.90, this also means that this threshold was not used (alone) for LEPAD definition and the inclusion of participants with LEPAD and IC. Unless all the participants in the IC group didn't have a LEPAD? This is quite puzzling, and authors should specify the criteria used to include participants with LEPAD.

In line with comment #1, the definition of IC should be then clearly outlined in the methods section. Further, the tool used for IC assessment is unknown.

7.      When were participants included? (i.e., when this study was conducted?)

8.      Line 108. Please, specify the model of activPAL used.

9.      Lines 119-130. Since the distance walked is an outcome measure, the GPS accuracy in distance estimation should be also reported and not only positional accuracy. Accordingly, the method used to compute walking distance should be also outlined since different methods exist.

Considering how the device was worn (line 145), the effect of wear location on GPS static and dynamic accuracy should be also outlined.

10.   **Data processing. There are very few information regarding the methods implemented to determine wear and non-wear periods, time in bed periods and thus valid time for analysis. In lines 236-239 (statistical analysis subsection?), we learn that a diary was used but this remains too elusive, and this should be outlined earlier. Further, did the authors used the CREA or VANE classification algorithm?

11.   Line 214. What was the minimal duration to define a walking event? (Which should define the minimal walking duration that could be detected)

12.   Line 243. Is the reference #40 directly related to the study of types of activity space used?

13.   ***Statistical analysis, lines 247-259. The authors used “a paired statistical approach” and compared outcome measures between IC and control by comparing pairs of matched IC and control participants, and thus used a paired t-tests. The use of a paired t-test to compare two independent groups isn’t a suitable statistical approach if we consider that one of the assumptions of a paired t-test is that the observations come from the same group of participants. Further, although the authors were interested by the study of two factors (IC and location), they analysed those two factors separately by multiplying the number of (paired) t-tests as a function of the number of comparisons assessed. This approach is known to increase Type 1 error, and no adjustment of the p-value was reported. By studying two factors, I wonder why the authors didn’t use a two-way ANOVA, together with an appropriate post-hoc test.

Further, is a Shapiro-Wilk test no more suitable than a Kolgorov-Smirnov test, considering the study sample size?

14.   No information is provided on the sample size calculation or target (considering the hypothesis reported at the of the Introduction section).

Results section

15.   Lines 269-270. Mean self-reported BMI. Did participants know their own BMI? Or, did they report their size and mass, and then you have computed their BMI?

16.   Lines 274-275. Please, consider using ABI instead of ABPI. The mean and standard deviation of ABI should be reported.

17.   **Throughout the results section, the results for Home/away and Indoor/outdoor are studied separately, although those four location categories are not independent. The rationale for such a choice has not been clearly stated in the methods section. Further, it would be of interest to know at least the time spent (min, %) over the collection period in each of the following category: Home-indoor/Home-outdoor/Away-indoor/Away-outdoor.

18.   *** As reported by the authors, across the whole assessment period, individual with IC walked less (lower walking events), taking fewer steps and walking during a lower duration. Consequently, when the authors then compared IC and controls across the four location categories (Tables 1 and 2, mainly), it is not clear if the differences reported cannot only be explained by the fact that overall IC participants walked less than controls. Did the differences reported remain when adjusting for the whole volume of the studied outcome?

Discussion section

19.   Line 359. From which outcome the authors could state that both groups did not meet the physical activity recommendations?

20.   *Lines 372-374. The authors should outline how such inference can be supported? Did the authors assess if participants were car owners, or what their preferred transportation mode(s) was.were? Do the authors think this could also influence the locus of activity (independently of the presence of IC)?

21.   Lines 378-381. In the comparisons done with previous studies, consider that the differences observed could be also explained by the different monitors used.

22.   Line 390. Which study? Please specify the reference.

23.   Using wearable monitors, a previous study quantified walking pain manifestations during daily life walking in people with LEPAD (PMID: 31271680). Authors should consider whether the findings of this study could strengthen the discussion of their results.

24.   *Previous studies have used GPS to assess transportation modes. Do the authors consider that such assessment would have been benefit to their study and would have strengthen their findings? Since the activPAL algorithm classification (CREA) is expected to detect “seated transport”, did the authors investigate potential association between the locus of activity and time in seated transport obtained from the activPAL?

25.   *Lines 400-401. Is this not only explained by the total lower volume of walking overall? (see comments #18)

26.   Line 477-478. Not sure to understand what is an “optimal walking distance” and what is an “optimal quality of life”.

27.   Lines 471-479. Are such recommendations directly inferred from the results of the present study?

Reviewer 2 Report

Dear Authors,

I received a manuscript to be reviewed titled: „ Walking behaviour of individuals with Intermittent Claudication compared to matched controls: does location matter?“

I have several reservations concerning the work

Line 39 in the Introduction: you cite source 10 in the text, but the whole sentence does not make sense in the context of your text: "compared with matched control groups"

In line 41 you are paraphrasing the author of source 12 and ,among others, you refer to "poor walking capacity". What does that mean in your understanding?

Please consider re-explaining the abbreviation IC in subsection 2.1. Participants (line 84)

Please consider formulating inclusion and exclusion criteria, at least for the experimental group, that existence of which you currently only mention in the text. This would provide more clarity (lines 87-94). Include these criteria.

I wanted to go through the processing of the data, which according to the authors is provided elsewhere (ref. 32), but that source is not available.

I suggest reducing chapter 2.4 (including subchapters 2.4.1, 2.4.2) a lot of the facts in these subchapters are apparently in ref.32, but the main reason is that the information in the subchapters themselves is repetitive. If necessary, you can put them in the appendices

In subchapter 2.4.3, lines 206-209, I recommend removing the explanations in parentheses, it is sufficient to modify the first sentence stating that it is walk.... On the contrary, I am not clear about the explanation "away from home outdoors (all walking events categorised as both away from home and outdoors)". I am referring to the previous explanation that you are recording "walking at home"," away from home", "indoors" and "outdoors". Is this meant to be a combination of walking "away from home" and "outdoor"?  Is that not redundant in that case? Moreover, I did not find information about this activity anywhere in the results. I suggest that the item be deleted. Furthermore, please clarify if "indoor" also included walking at home, and also if "outdoor" included walking away from home.

We learn about "The mean Ankle Brachial Pressure Index (ABPI)"(line 274) for the first time only in the Results. In general, this does not matter, but in the given context, I wonder whether it is necessary to mention the above acronym if it is only mentioned twice in the whole text (first explained in line 274 and one more time in the Discussion, line 389).

Inconsistency in References:

The authors in line 628 are improperly listed under no. 32 (Iveson et al., 2020 + Wing et al., 2005), you thus have a duplicate position in References

Please double-check References. Some doi are not found (e.g. ref. 3, 4, 32)

Or they do not match the author (ref. 2 - listed as Leng et al., 1993; after opening the doi, the author is Horváth et al, 2022)

while only the first 4 references and the authors' previous article have been checked

Author Response

Please see attached file.  Responses to reviewer 2 comments are towards the end  

Round 2

Reviewer 1 Report

Thank you to the authors for their responses to my previous comments and for the work done, which – to my opinion - significantly improved the manuscript. I have only two last comments below. Thank you.

Response #3

Again, when the authors state that “Home-based exercise programs, (which) provides structured advice to increase physical activity (PA) […]”, to my opinion this assertion isn't correct. Home-based exercise programs first provide recommendations on structured walking exercise sessions to increase walking ability and decrease lower-limbs symptoms, and do not primarily focus on advice to change PA activity. Unless the authors could warrant it with appropriate references, the sentence should be properly aligned with the scientific knowledge.

Response #6

Thank you for those clarifications. I would suggest keeping the ABI values to better describe the studied population. When appropriate throughout the manuscript, it would be important to clarify that the studied participants are participants with a "suspected" LEPAD. Further the issue of appropriately defining the population should be add as a limitation. Again, I’m not sure that a clear definition of IC was provided.

Author Response

Thank you to the authors for their responses to my previous comments and for the work done, which – to my opinion - significantly improved the manuscript. I have only two last comments below. Thank you.

Authors response

Thank you again to the reviewers for the opportunity to amend the manuscript.  We accepted all previous tracked changes, and have added changes from this round of reviewers as tracked changes.  A response to each point is made below.  We added one reference to the article, and have updated the reference numbers accordingly.

Response #3

Again, when the authors state that “Home-based exercise programs, (which) provides structured advice to increase physical activity (PA) […]”, to my opinion this assertion isn't correct. Home-based exercise programs first provide recommendations on structured walking exercise sessions to increase walking ability and decrease lower-limbs symptoms, and do not primarily focus on advice to change PA activity. Unless the authors could warrant it with appropriate references, the sentence should be properly aligned with the scientific knowledge.

Author Response:

We were basing our definition on that used in the protocol for the systematic review of Pymer et al 2021 (reference [7]), which stated their definition HEP to be included in the review was “The HEP intervention will include structured advice to increase physical activity by guiding patients in terms of frequency, intensity and/or duration rather than basic advice to ‘go home and walk’”. We therefore feel the definition can include instructions to increase physical activity. However, we do agree with the reviewer, that such instructions would be anticipated to be structured, and are not necessarily intended to increase physical activity. We have therefore amended our text in lines [36-38]:

“Home-based exercise programmes, which provides structured advice on walking exercise and may include advice to increase physical activity (PA), often alongside other behavioural components, may offer a more accessible and preferable alternative to supervised exercise therapy”

Response #6

Thank you for those clarifications. I would suggest keeping the ABI values to better describe the studied population. When appropriate throughout the manuscript, it would be important to clarify that the studied participants are participants with a "suspected" LEPAD. Further the issue of appropriately defining the population should be add as a limitation. Again, I’m not sure that a clear definition of IC was provided.

Author response:

Thank you for the suggestions.  We have re-instated the ABI results, and also reported the Lothian Oximetry Index results when they were reported as an alternative [Lines 298-301].

The mean ABI of participants with IC was 0.71 (n=20; range: 0.54-1.12) and the LOI was 0.78 (n=7; range: 0.58-1.14), data was missing for one participant. As a whole, the group with IC had moderate disease severity, but it should be noted that some ABI (n=3) and LOI (n=1) values were higher than 0.9

We have therefore also added a section to the methods reporting how we obtained these values from medical records [Lines 159-169].

Information on disease severity was obtained from medical notes rather than being specifically measured for the research study. All assessment of peripheral blood flow were taken by specialist staff working in a vascular screening clinic. In the clinics, disease severity could be evaluated using various metrics. Typically, the ankle-brachial index (ABI) was employed, which is the ratio of ankle to brachial systolic blood pressure. This measurement was taken while the participant was supine, either using a handheld Doppler device or machines provide simultaneous readings. Where patients had diabetes or non-compression of the vessels were suspected, the Lanarkshire oximetry index (LOI) was employed, which uses a pulse oximeter at the toe and finger as an alternative to a handheld Doppler device [41]. ABI and LOI are interpreted similarly, with a value of ≤0.9 considered indicative of PAD [5].

And we have added a section into the limitations section regarding this issue [Lines 497-505]. 

Participants in the IC group had received a diagnosis of IC from the NHS specialist vascular clinics which they were attending, but we did not specifically measure any indicators of PAD or IC for this research project. Some of the measures of disease severity we obtained from medical records (ABI and LOI) indicated values which were in the normal range (>1.0). Equally, although we screened for symptoms of IC in the control group, it is possible that some individuals in that group had asymptomatic PAD. However, any misallocation to group would be anticipated to act in a conservative manner to reduce differences between groups.

Regarding changing language surrounding the diagnosis/suspicion of PAD.  We do not feel comfortable with this suggestion.  We recruited individuals who had come through specialist vascular clinics in the UK health system, and our stance is that information of diagnosis from these clinics is credible.  We have clarified for the international reader the nature of the clinics within the UK health service [Lines 102-103 and Line 498].

“… returning patients from the Community Claudication Clinics Service (a specialist vascular screening service) across NHS Lanarkshire, UK.”

“… from the NHS specialist vascular clinics which they were attending …”

Regarding definitions of IC.  In the original text of the introduction we provided a description of IC [Lines 28-32] as:

“Intermittent Claudication (IC) is considered a key symptom, in which individuals experience cramp-like pain in their lower limb(s) during walking due to insufficient lower limb circulation to meet the metabolic demands of the muscles”.

We have updated the description of IC in the methods to include some additional detail [Lines 104-107]

“Individuals were included in the study if they had a medical diagnosis of IC from the claudication clinic (based on the Edinburgh Claudication Questionnaire [29], confirming that pain was associated with walking and resolved with rest).”

Reviewer 2 Report

Thank you for respecting my comments and including them in the article. In this form, the article is better and clearer and can be republished from my point of view.

Author Response

thank you

Round 3

Reviewer 1 Report

Thank you to the authors for this final round of modifications and congratulations for their study.